# Iron Beats Electricity: Resistance Training but Not Whole-Body Electromyostimulation Improves Cardiometabolic Health in Obese Metabolic Syndrome Patients during Caloric Restriction—A Randomized-Controlled Study

**DOI:** 10.3390/nu13051640

**Published:** 2021-05-13

**Authors:** Dejan Reljic, Hans J. Herrmann, Markus F. Neurath, Yurdagül Zopf

**Affiliations:** 1Hector-Center for Nutrition, Exercise and Sports, Department of Medicine 1, University Hospital Erlangen, Friedrich-Alexander University Erlangen-Nürnberg, 91054 Erlangen, Germany; hans.herrmann@uk-erlangen.de (H.J.H.); yurdaguel.zopf@uk-erlangen.de (Y.Z.); 2German Center Immunotherapy (DZI), University Hospital Erlangen, Friedrich-Alexander University Erlangen-Nürnberg, 91054 Erlangen, Germany; markus.neurath@uk-erlangen.de; 3Department of Medicine 1, University Hospital Erlangen, Friedrich-Alexander University Erlangen-Nürnberg, 91054 Erlangen, Germany

**Keywords:** exercise, obesity, fasting, muscle mass, strength training, cardiovascular disease, hypertension, electrical stimulation, muscular strength, psychological health

## Abstract

Caloric restriction (CR) and exercise are cornerstones in the treatment of obesity and cardiometabolic disorders. Recently, whole body electromyostimulation (WB-EMS) has emerged as a more time-efficient alternative to traditional resistance training (RT). However, the effects of WB-EMS compared to RT on cardiometabolic health in obese metabolic syndrome (MetS) patients performed during CR are still unclear. In total, 118 obese MetS patients (52.7 ± 11.8 years, BMI: 38.1 ± 6.9 kg/m^2^) undergoing CR over 12 weeks (aim: −500 kcal deficit/day) were randomly allocated to either WB-EMS, single-set RT (1-RT), 3-set RT (3-RT) or an inactive control group (CON). Primary outcome was MetS severity (MetS z-score). Secondary outcomes were body composition, muscle strength and quality of life (QoL). All groups significantly reduced body weight (~3%) and fat mass (~2.6 kg) but only 1-RT and 3-RT preserved skeletal muscle mass (SMM). All exercise groups increased muscle strength in major muscle groups (20–103%). However, only the two RT-groups improved MetS z-score (1-RT: −1.34, *p* = 0.003; 3-RT: −2.06, *p* < 0.001) and QoL (1-RT: +6%, *p* = 0.027; 3-RT: +12%, *p* < 0.001), while WB-EMS and CON had no impact on these outcomes. We conclude that traditional RT has superior effects on cardiometabolic health, SMM and QoL in obese MetS patients undergoing CR than WB-EMS.

## 1. Introduction

The global prevalence rates of obesity have continued to rise over the last decades [1]. Increased body weight with an excessive accumulation of body fat is associated with an elevated risk of a number of chronic diseases, such as cardiovascular disease (CVD), type 2 diabetes and some cancer types [2]. The clustering of obesity-related cardiometabolic risk factors, including hypertension, excess visceral fat, abnormal blood lipid, and glucose levels—also known as the metabolic syndrome (MetS)—additionally raises the risk of serious health problems and premature mortality [3]. Moreover, it has been observed during the current COVID-19 pandemic that patients with obesity and preexisting cardiometabolic conditions are at a substantially higher risk of developing a severe clinical course following a coronavirus infection [4,5]. Thus, the development and evaluation of viable approaches to treat obesity and related cardiometabolic conditions is probably more pertinent than ever.

Diet modification, in particular caloric restriction (CR), is a crucial component in the treatment of obesity [6,7]. It has been suggested that both intermittent and continuous CR may provide beneficial effects in promoting weight loss and metabolic improvements in overweight/obese men and women [8]. However, weight loss through CR alone does not only affect fat mass (FM) but is typically also accompanied by a significant decrease in skeletal muscle mass (SMM) and other body tissues [9,10]. Loss of SMM may be an unfavorable side effect of CR for several reasons [11]. It has been documented, for example, that a reduction in SMM translates into higher prevalence of musculoskeletal disorders, diminished ability to perform activities of daily living and increased risk of injuries [12]. Given that SMM is the largest metabolically active tissue in the body with significant impact on glucose regulation, insulin sensitivity and resting metabolic rate [13,14], muscle wasting can also have an adverse impact on cardiometabolic health. Furthermore, it has been shown that CR may cause negative effects on cardiorespiratory fitness, possibly due to detrimental changes in the cardiovascular system [10,15], and processes involved in hematopoiesis [16].

Therefore, physical exercise is advocated as an integral part of a comprehensive obesity treatment program [9,17,18]. Exercise not only contributes to the energy deficit required for weight loss and maintenance but may also prevent some of the undesirable effects of CR on SMM [9,17] and aerobic capacity [10]. In addition, it has been reported that exercise and a higher level of physical fitness improve risk factors that underlie MetS, regardless of body mass index (BMI) [19]—even in the absence of significant weight loss [20]. Besides aerobic exercise, which primarily aims to improve cardiorespiratory fitness [19], it is recommended that resistance training (RT) should be included in a well-rounded exercise program in order to counteract a loss of SMM and muscle strength [17,18].

However, despite vast evidence for the important role of regular exercise in the treatment of obesity and related disorders, the majority of obese individuals are not reaching the recommended physical activity levels (i.e., a minimum of 150 min of moderate or 75 min of higher intensity aerobic physical activity plus muscle-strengthening exercises on two or more days each week [21]), with lack of time being the most commonly cited barrier to adopting and maintaining regular exercise [22,23]. Moreover, it has been suggested that presenting a “150 min per week threshold” as a necessary amount for achieving health benefits through physical activity may have adverse impact on motivation because it is not a realistic goal for most individuals [24]. Accordingly, the development of more time-efficient exercise strategies for improving health outcomes has increasingly moved into the focus of research. Recent studies have shown, for example, that low-volume high-intensity interval training (HIIT), a particularly time-efficient type of cardiovascular training, can significantly enhance cardiometabolic risk profile in obese patients with MetS, despite considerably lower time effort compared to conventional aerobic exercise modalities [25,26].

Regarding muscle strengthening exercise, whole-body electromyostimulation (WB-EMS) has recently emerged as an innovative training approach that is advertised as a more time-efficient alternative to conventional RT [27]. Compared to the local electromyostimulation method, originally developed to address single muscles, the novel WB-EMS technology allows a simultaneous activation of all major muscle groups through the application of an electrical current by means of specific exercise suits with built-in electrodes. To date, most studies that have evaluated the efficacy of WB-EMS have included athletes, moderately trained or sedentary healthy subjects [27,28,29], while data on the application in obese populations are still limited. First studies have shown that WB-EMS can improve body composition, strength, and certain cardiometabolic risk indices in premenopausal overweight women [30] and sarcopenic obese but otherwise healthy older adults [31,32,33]. More recently, a pilot study from our group revealed that WB-EMS may beneficially affect muscle strength and MetS severity in obese female MetS patients [34]. However, more research is needed before these preliminary results can be generalized on obese populations at increased cardiometabolic risk. Moreover, surprisingly, only one study has directly compared the effects of WB-EMS versus traditional RT. In this study, Kemmler et al. found that both exercise modalities induced similar improvements in body composition, strength [35] and cardiometabolic risk outcomes [36] in healthy untrained men. However, to date, corresponding comparative studies in cardiometabolic risk groups are still lacking, and thus, it is currently unclear whether the two types of exercise confer similar effects in obese MetS patients undergoing CR.

Therefore, the purpose of the present study was twofold: First, we aimed to verify the previous findings obtained in our pilot study about the impact of WB-EMS on cardiometabolic health in a larger sample of obese MetS patients. Second, we aimed to compare the effects of WB-EMS versus time-matched single-set RT (1-RT), standard 3-set RT (3-RT) or no exercise (CON) during a 12-week period of CR on MetS severity, body composition, muscle strength and quality of life (QoL) in obese patients with MetS. We hypothesized that (i) 3-RT would have the greatest beneficial impact on MetS severity, SMM and muscle strength, (ii) WB-EMS and 1-RT would have similar effects on these outcomes, as observed in previous research in healthy untrained men [35,36], and (iii) all three exercise modalities would provide greater benefits compared to CR without additional exercise.

## 2. Materials and Methods

### 2.1. Study Design

This study was a randomized-controlled trial over a period of 12 weeks, involving an exercise intervention, nutritional counseling and pre-/post-intervention health examinations. Participants were allocated in a random manner to one of three exercise groups (either performing WB-EMS, 1-RT or 3-RT) plus nutritional counseling, or an inactive CON group, only receiving nutritional counseling. Group allocation was performed using a computer-generated random number sequence (MinimPy, GNU GPL v3) by a researcher who was not involved in data collection. Primary outcome was the metabolic syndrome z-score (MetS z-score), a continuous score of the five MetS variables as further specified in the Methods section, and secondary outcomes were body composition (in particular SMM and FM), maximum strength (i.e., one repetition maximum, 1RM) of the major muscle groups and self-reported QoL. Participants were completely informed about the studies’ aims and procedures, which conformed to the Helsinki Declaration, and provided written consent before enrolment. The study protocol was approved by the Ethical Committee of the Medical Faculty of the Friedrich-Alexander University Erlangen-Nürnberg (approval number: 203_17B) and registered at ClinicalTrials.gov (ID-number: NCT03306056).

### 2.2. Participants

Participants were recruited through local newspaper advertisements and flyers posted in medical practices. Eligibility criteria for study participation were: age ≥ 18 years, obesity (BMI ≥ 30 kg/m^2^) and increased waist circumference (>88 cm for females, >102 cm for males) plus a minimum of two further cardiometabolic risk factors, including hypertension (≥130 mmHg systolic and/or ≥85 mmHg diastolic blood pressure), dyslipidemia (triglycerides: ≥150 mg/dL; high-density lipoprotein cholesterol (HDL-C)): <50 mg/dL for females, <40 mg/dL for males) and hyperglycemia (≥100 mg/dL) [37], and a self-reported sedentary lifestyle as defined previously [38]. Exclusion criteria were the following: clinical diagnosis of heart disease, cancer, severe orthopedic disorders or other health conditions that might preclude safe participation in an exercise program and pregnancy. Participants agreed not to change their medications/supplements or dosages without medical consultation and informing the principal investigator and to maintain their usual physical activity habits throughout the study to minimize potential confounding effects. On the basis of previous data [34], indicating a moderate to large effect (d = 0.63) on the primary outcome (MetS z-score), a priori sample size calculation suggested that 16 participants per group would be needed to achieve a 95% power in a 2-way repeated measures ANOVA with four groups and two measuring times at a 5% level of significance (G*Power, version 3.1.9.2, Heinrich Heine University Düsseldorf, Germany). To sufficiently account for dropouts, which have been reported to range between 30–80% in obesity interventions [39], we aimed to recruit 30 participants per group.

### 2.3. Health Examination

One week before the start of the intervention, participants completed the baseline examination, which also included resting and exercise electrocardiography to ensure a safe participation in the exercise program. The second examination was performed within the first week after completion of the intervention, at least 3 days apart from the final exercise session and at a similar time of day to ensure adequate recovery and to avoid potential circadian effects. All procedures were conducted under laboratory conditions at our Research Center and were strictly standardized as outlined below. Participants were instructed to arrive after an overnight fast and to refrain from alcohol and intense physical activity for at least 24 h prior to the examination day. All examinations were made in a single-blinded fashion, meaning that the researchers who were involved in data collection were unaware of the participants’ group allocation.

#### 2.3.1. Blood Pressure Measurements

Participants were first advised to empty their bladder and to rest in a sitting position for 5 min. Subsequently, systolic and diastolic blood pressure values were measured by means of an automatic upper-arm blood pressure monitor (M5 professional, Omron, Mannheim, Germany), which has been validated for accuracy [40]. As recommended by guidelines [41], two consecutive measurements on both arms were obtained at 60 s intervals, and the mean values of the arm with the higher pressure were used in the analysis.

#### 2.3.2. Blood Collection

Blood samples were drawn via venipuncture from an antecubital arm vein and analyzed at the diagnostic laboratories of the University Hospital Erlangen. Serum concentrations of glucose, triglycerides, total cholesterol, low-density lipoprotein cholesterol (LDL-C), and high-density lipoprotein cholesterol (HDL-C) were determined photometrically (Clinical Chemistry Analyzer AU700/5800, Beckman Coulter, Brea, CA, USA). Serum glycosylated hemoglobin A1c (HbA1c) was measured using a turbidimetric immunoassay (COBAS Integra 400, Roche Diagnostics, Mannheim, Germany). For safety reasons, creatine kinase (CK) and creatinine levels (Clinical Chemistry Analyzer AU700/5800, Beckman Coulter, Brea, CA, USA) were assessed baseline, at week 6 and week 12, to monitor muscular stress and potential effects on renal function.

#### 2.3.3. Assessment of Body Composition

Body composition was assessed using a segmental multi-frequency bioelectrical impedance analysis (BIA) device (seca mBCA 515, Seca, Hamburg, Germany). The device has been validated against magnetic resonance imaging (MRI), the gold standard for the measurement of SMM [42], and shown to provide precise body composition measures in obese individuals when compared to the 4-compartment reference method [43]. Waist circumference (WC) was measured in a standing position with a non-stretchable measuring tape.

#### 2.3.4. Determination of MetS z-Score

MetS z-score was based on WC, mean arterial blood pressure (MAB), serum concentrations of fasting serum glucose (GLU), triglycerides (TG), and HDL-C and calculated according to the following sex-specific (males = M, females = F) equations [44]:M: [(40 − HDL-C)/9.0] + [(TG − 150)/81.0] + [(FBG − 100)/11.3] + [(WC − 102)/7.7] + [(MAB − 100)/9.1]
F: [(50 − HDL-C)/14.1] + [(TG − 150)/81.0] + [(GLU − 100)/11.3] + [(WC − 88)/9.0] + [(MAB − 100)/9.1]

#### 2.3.5. Maximum Strength (1RM) Testing

1RM of the major muscle groups (abdominals, lower back, upper back, chest, and legs) was estimated baseline, at week 5, week 9, and post-intervention through submaximal tests, based on the performance of multiple repetitions. Compared to the common 1RM test (i.e., the maximum weight an individual can lift for one repetition), which is considered the gold standard for assessing maximum strength in athletes and trained individuals [45], multiple repetition tests involve less risk to induce injuries of the musculoskeletal system and are therefore recommended for previously untrained individuals [46,47,48]. After a brief warm-up, initial instructions, and familiarization with the test procedures, participants performed 1RM tests on the following five devices in a standardized order: Chest press, lat pulldown machine, lower back machine, abdominal crunch, and leg press (TechnoGym, Neu-Isenburg, Germany). Testing was supervised by certified physiotherapists/sports therapists. Initial load was estimated based on the therapist’s experience and participants’ feedback on the chosen weight load. Subsequently, participants were instructed to perform as many repetitions as possible with the given load until concentric movement failure. The number of repetitions was not supposed to exceed six repetitions, which has been shown to be an appropriate number to accurately predict 1RM [46]. If the repetition number was more than six, the load was increased and a new attempt was performed after 3 min of rest. The detection of the respective weight was typically achieved within a maximum of three attempts. 1RM was calculated according to the Brzycki equation [49], as follows:1RM = 100 × load rep/(102.78 − 2.78 × rep)

#### 2.3.6. Assessment of Self-Reported Outcomes

Health-related QoL was determined by the EQ-5D-5L questionnaire, consisting of a visual analogue scale (EQ-VAS, 0–100 points, higher values correspond to higher QoL) and a descriptive system of five health-related QoL-dimensions (mobility, self-care, usual activities, pain/discomfort, anxiety/depression, each with five severity levels) transformed to a single index score (EQ-5D-5L). A score of 1.0 marks the best possible state of perceived health, while a score of 0 marks the worst possible health state. The questionnaire was previously validated in a German population [50]. In addition, participants allocated to one of the three exercise groups provided a personal evaluation sheet after termination of the intervention period to rate how much they enjoyed the exercise program on a 7-point rating scale (1 = not enjoyable at all; 7 = extremely enjoyable).

### 2.4. Nutritional Counseling

Participants received nutritional counseling by a registered dietitian in a face-to-face meeting at study enrolment. As recommended by international guidelines for the treatment of obesity [6], participants were instructed to reach an energy deficit of 500 kcal per day while maintaining proper protein consumption (i.e., ≥1.0 g/kg/day) to support SMM preservation during CR [51]. Additionally, meal planning handouts and food list with corresponding calorie amounts were provided to support participants on how to implement the dietary recommendations at home. Additional consultation was offered by our nutrition team throughout the whole study period if questions arose regarding the diet. Nutritional intakes were controlled using 24 h dietary records (Freiburger Ernährungsprotokoll; Nutri-Science, Freiburg, Germany) assessed on three consecutive days at the onset of the study and within the last intervention week. Mean caloric and nutrient intake was analyzed using the software PRODI 6 expert (Nutri-Science, Freiburg, Germany).

### 2.5. Exercise Training Programs

All exercise sessions were supervised one-to-one by certified physiotherapist/sports therapists, who were trained in implementing the specific training protocol. Participants had the opportunity to arrange their exercise sessions individually during the opening hours of the training center. Each session commenced with a brief 5 min warm-up at low intensity on a cycle ergometer. Training was performed twice weekly with at least two days rest between sessions over a period of 12 weeks (a total of 24 sessions).

#### 2.5.1. Whole-Body Electromyostimulation

WB-EMS exercise was performed using devices and equipment from miha bodytec (Gersthofen, Germany), including a vest, a hip belt and upper-arm and thigh cuffs with integrated electrodes to induce the electrical muscle stimulation. Electrical muscle stimulation was applied by bipolar impulses at a frequency of 85 Hz and a pulse width of 350 μs inducing an intermittent stimulation of a 6 s impulse phase followed by 4 s rest as previously described and most frequently used in commercial fitness settings [27]. In total, eight muscle groups were addressed by the WB-EMS application, namely, upper arms, chest, upper back, latissimus, abdomen, lower back, buttocks, and thighs. Current intensity was set to trigger a noticeable muscle contraction. During the impulse phase, participants performed two sets of light dynamic movements, each repeated for ten times, including trunk flexion and extension, moderate squats, butterfly movements, and lat pull-down movements to support the activation of the mentioned muscle groups (total session time: 20 min). Current intensity was individually adapted in each session and, where appropriate, increased accordingly to ensure training load progression.

#### 2.5.2. Resistance Training

RT consisted of five machine-supported exercises to target all major muscle groups (abdominals, lower back, upper back, chest, and legs) using the following exercise devices: Chest press, lat pulldown machine, lower back machine, abdominal crunch, and leg press (TechnoGym, Neu-Isenburg, Germany). Based on the 1RM tests performed every 4 weeks, the weight load was progressively increased over the 12 weeks to achieve the following target ranges: week 1–4: 50–60% 1RM; week 5–8: 60–75% 1RM; week 9–12: 70–80% 1RM. Repetitions were performed with 2 s of concentric (weight lifting phase) and 2 s of eccentric (weight lowering phase) muscle work until failure. The 1-RT group performed one set of each exercise with a resting period of 2 min between exercises. The 3-RT consisted of three sets of each exercise with similar resting periods between sets and exercises as 1-RT. As the number of repetitions decreased with increasing intensity, the mean session time decreased during the training period from ~20 min/session to ~11 min/session in the 1-RT group and from ~60 min/session to ~38 min/session in the 3-RT group, respectively.

### 2.6. Statistical Analysis

All analyses were performed using SPSS version 24.0 (SPSS Inc., Chicago, IL, USA). Initially, the Shapiro–Wilk test was applied to check the distribution of data. A 2 × 2 repeated-measures ANOVA was performed to determine main effects of group, time, and interaction between both factors. Additionally, gender subgroup analyses were performed for the primary outcome MetS z-score. Homogeneity of variance was assessed by the Levene’s test. In case of significant main or interaction effects, post hoc paired t-tests or 1-way ANOVAs followed by Sidak’s post hoc tests were carried out to determine within-group changes and between-group differences, respectively. In case of non-normally distributed data, log or square root transformation was used, and subsequently, the same analyses were applied to the transformed values. If data could not be transformed to meet normalization, non-parametric Friedman two-way analysis of variance by ranks was applied, followed by Wilcoxon’s and Mann–Whitney tests for post hoc comparisons. Effect sizes were determined according to the partial eta-squared (η*p*^2^) for ANOVA and Kendall’s coefficient of concordance (*W*) for the Friedman test. Effect sizes were classified as: small ≤ 0.01, medium ≥ 0.06, and large ≥ 0.14 for η*p*^2^, and small ≤ 0.10, medium ≥ 0.30, and large ≥ 0.50 for *W* [52]. For all analyses, the significance level was defined to be *p* < 0.05. Data are reported as means ± standard deviation (SD) and pre-/post-intervention changes are shown with 95% confidence intervals (95% CI).

## 3. Results

### 3.1. Study Flow

A total of 130 individuals were screened for eligibility. All participants had already experienced multiple weight loss attempts in the past. Five individuals were excluded for not meeting the inclusion criteria. After the baseline examination, four individuals withdrew for personal reasons, two were excluded after the health examination due to medical reasons, and one could not be included in the study due to an unrelated injury. Thus, 118 participants were randomized to either (i) whole-body electromyostimulation (WB-EMS, *n* = 31), (ii) 1-set resistance training (1-RT, *n* = 28), (iii) 3-set resistance training (3-RT, *n* = 29), or (iv) an inactive control group (CON, *n* = 30). Table 1 shows the participants’ main baseline characteristics within each group. During the intervention period, 28 participants dropped out (WB-EMS = 8, 1-RT = 5, 3-RT = 6, CON = 8). The reasons for dropout are shown in Figure 1 (Study Flow Chart). A total of 91 participants completed the study and were included in the final analysis (WB-EMS: *n* = 23, 1-RT: *n* = 23, 3-RT: *n* = 23, CON: *n* = 22). The adherence rates (the percentage of the scheduled training sessions that were completed) were high in all exercise protocols (WB-EMS: 93.2 ± 7.8%, 1-RT: 93.6 ± 6.5%, and 3-RT: 94.7 ± 6.7%) with no significant differences between the three groups. There were no adverse events at any point during the training sessions. No significant effects of gender were found, and therefore, the results of both males and females were considered together in all analyses.

### 3.2. Nutritional Analysis

Significant main effects of time were found for energy (*p* < 0.001, *ή*^2^ = 0.30), protein (*p* < 0.001, *ή*^2^ = 0.22), fat (*p* < 0.001, *ή*^2^ = 0.18), and carbohydrate (*p* < 0.001, *ή*^2^ = 0.28) intakes. Subsequent post hoc tests revealed that the average total daily energy intake per day decreased significantly in all groups from baseline to follow-up (WB-EMS group: −497 ± 648 kcal, 95% CI: −800 to –194 kcal, *p* = 0.003; 1-RT: −505 ± 850 kcal, 95% CI: −892 to −118 kcal, *p* = 0.013; 3-RT: −516 ± 556 kcal, 95% CI: –769 to −262 kcal, *p* < 0.001; CON: −580 ± 1131 kcal, 95% CI: −1109 to −50 kcal, *p* = 0.034). There were no significant differences in caloric and macronutrient intakes between groups. Group-specific nutritional intakes are shown in Table 2.

### 3.3. Anthropometric Data and Body Composition

Significant main time effects were found for body weight (*p* < 0.001, W = 0.27), BMI (*p* < 0.001, *ή*^2^ = 0.38), FM (*p* < 0.001, *ή*^2^ = 0.40), %FM (*p* < 0.001, W = 0.19), SMM (*p* = 0.001, W = 0.06), body water (*p* = 0.009, W = 0.04), and WC (*p* < 0.001, ή^2^ = 0.21). A significant group-by-time interaction was observed for WC (*p* < 0.001, *ή*^2^ = 0.11). All groups achieved a significant reduction of body weight (WB-EMS: −4.0 kg, 95% CI: −6.0 to −2.1 kg, *p* < 0.001; 1-RT: −3.2 kg, 95% CI: −5.5 to −0.9 kg, *p* = 0.004; 3-RT: −3.3 kg, 95% CI: −5.1 to −1.6 kg, *p* = 0.004; CON: −2.5 kg, 95% CI: −3.7 to −1.2 kg, *p* < 0.001), mainly due to a decrease in FM (Figure 2c). The amount of weight loss did not differ significantly between groups (Figure 2a). However, reductions in WC were significantly greater in the 1-RT (*p* = 0.005) and 3-RT group (*p* = 0.004) compared to CON (Figure 2b). In the WB-EMS and CON group, there was a significant decrease of SMM, whereas in the 1-RT and 3-RT group, SMM remained stable (Figure 2d). Pre- and post-intervention values of each group are shown in Table 3.

### 3.4. Cardiometabolic Risk Markers

A significant main effect of time was detected for SBP (*p* < 0.001, *ή*^2^ = 0.10), DBP (*p* = 0.025, *ή*^2^ = 0.04), MAB (*p* < 0.001, *ή*^2^ = 0.09), HbA_1c_ (*p* = 0.002, *W* = 0.07), total cholesterol (*p* < 0.001, *W* = 0.12), HDL-C (*p* = 0.002, *W* = 0.07), LDL-C (*p* < 0.001, *ή*^2^ = 0.09) and MetS z-score (*p* < 0.001 *ή*^2^ = 0.19). A significant group-by-time interaction was found for DBP (*p* = 0.021, *ή*^2^ = 0.08) and MetS z-score (*p* = 0.019, *ή*^2^ = 0.09). Post hoc tests showed significant reductions of SBP and MAB in the 1-RT (−10 mmHg, 95% CI: −18 to −2 mmHg, *p* = 0.016, and −7 mmHg, 95% CI: −13 to −2 mmHg, *p* = 0.009, respectively) and 3-RT group (−10 mmHg, 95% CI: −16 to −4 mmHg, *p* = 0.001, and −6 mmHg, 95% CI: −10 to −2 mmHg, *p* = 0.002, respectively). In the 3-RT group, DBP was also significantly decreased post-intervention (−4 mmHg, 95% CI: −7 to −1 mmHg, *p* = 0.009). Reductions in MAB were significantly greater in the 1-RT (−9 mmHg, 95% CI: −17 to −2 mmHg, *p* = 0.009) and 3-RT group (−8 mmHg, 95% CI: −16 to −1 mmHg, *p* = 0.034) compared to the WB-EMS group. Post hoc tests did not reveal significant group specific changes in any blood marker, except for cholesterol and LDL-C in the 1-RT group, which were decreased by 10 mg/dL (95% CI: −18 to −1 mg/dL, *p* = 0.034) and 9 mg/dL (95% CI: −16 to −1 mg/dL, *p* = 0.024), respectively, and total cholesterol in the CON group (−11 mg/dL, 95% CI: −21 to −3 mg/dL, *p* = 0.012). Post-intervention MetS z-score was only found to be significantly reduced in the 1-RT (−1.34 units, 95% CI: −2.2 to −0.5 units, *p* = 0.003) and 3-RT group (−2.06 units, 95% CI: −3.0 to −1.2 units, *p* < 0.001). The reduction in MetS z-score tended to be larger 3-RT group compared to the WB-EMS group (*p* = 0.051) (Figure 3). Cardiometabolic risk markers in each group are shown in Table 4.

### 3.5. Muscle Strength

A significant main effect of time was observed for muscle strength in abdominals (*p* < 0.001, *W* = 0.33), lower back (*p* < 0.001, *W* = 0.39), upper back (*p* < 0.001, *W* = 0.37), chest (*p* < 0.001, *W* = 0.41) and legs (*p* < 0.001, *W* = 0.27). All three exercise modalities significantly improved strength performance in the assessed muscle groups from pre- to post-intervention, whereas no significant changes were observed in the CON group (Table 5). The 3-RT group achieved significantly greater pre- to post-intervention strength gains in abdominal muscles compared to WB-EMS (Figure 4). Pre- and post-intervention values are shown in Table 5, and strength progress over time for each group is displayed in Figure 4.

### 3.6. Blood Markers of Muscle Status and Renal Function

There were no significant within- and between-group differences in blood markers of muscle status and renal function (Table 6).

### 3.7. Self-Reported Outcomes

Self-reported QoL changed significantly over time (*p* < 0.001, *W* = 0.19). Group-specific analyses revealed significant increases on the EQ VAS scale in the 1-RT (+6%, 95% CI: 1% to 10%, *p* = 0.027) and 3-RT group (+12%, 95% CI: 6% to 11%, *p* < 0.001). No significant changes were found in the WB-EMS and CON group. Ratings of exercise enjoyment were high among the three exercise groups, without significant between-group differences. A high proportion of the participants allocated to the exercise groups (~90%) stated that they intended to continue engaging in their exercise protocol after completion of the study. Group-specific values are presented in Table 7.

## 4. Discussion

The key findings of this study were the following: (i) A modest average body weight reduction of ~3% from CR resulted in a significant loss of SMM in obese MetS patients. (ii) Concomitant muscle-strengthening exercise during CR, either performed as traditional resistance training (1-RT, 3-RT) or WB-EMS, increased muscle strength in major muscle groups; however, only 1-RT and 3-RT were effective to prevent a significant loss of SMM. (iii) Most importantly, only the two conventional RT programs led to improvements in MetS severity and health-related QoL, whereas WB-EMS and CR without additional exercise (CON) did not have any significant impact on these outcomes.

The significant loss of SMM during the 12-week period of CR in the inactive CON group was not an unprecedented finding. It is well documented in the literature that the catabolic state induced by longer-term CR can lead to increased muscle wasting [9,10,11], and it is not uncommon that several weeks of reduced energy intake may result in a significant reduction of lean body mass of up to 1 kg or more [10]. Although it has been shown that CR per se may contribute to improve cardiometabolic risk outcomes [8], there is also strong evidence that loss of SMM is independently associated with diminished cardiometabolic health [53] and impaired physical performance [10]. Accordingly, the CON group did not experience significant improvements in overall cardiometabolic health status and muscle strength the in post-intervention examinations. This finding reinforces long-standing recommendations that obesity treatments should target changes in body composition, cardiometabolic risk indices and physical performance rather than pure weight loss [20]. In this regard, it has been shown that RT can serve as a potent countermeasure to prevent loss of SMM and muscle strength during CR-induced weight loss [11,54]. Furthermore, there is a plethora of evidence supporting the benefits of regular RT on cardiometabolic risk outcomes in both healthy and clinical populations [55]. In accordance with the literature [55], we observed significant improvements in cardiometabolic health status in our participants performing RT, which underpins the beneficial role of muscle strengthening exercise during CR in obese individuals at increased cardiometabolic risk. As expected, 3-RT had the most favorable effects on muscle strength and the cardiometabolic risk profile among the three exercise groups. It is to note, however, that participants from the 1-RT group experienced almost similar improvements in most cardiometabolic risk factors as the 3-RT group, despite substantially lower exercise volume and time commitment. Although still controversially discussed [56], this finding supports previous research, reporting that single-set RT programs may initially produce most of the health benefits of multiple-set RT programs in previously untrained individuals and clinical populations [54,55,57]. Thus, our data indicate that 1-RT can be regarded as an effective and time-efficient exercise option for obese MetS patients who are not able or willing to engage in more time-consuming multiple-set RT programs—at least in the early phases of an exercise program.

During the last decade, WB-EMS has gained growing popularity in commercial fitness facilities, where it is increasingly offered as a time-saving alternative to traditional multiple-set RT. Although it has recently been pointed out that scientific data on the effectiveness of this rather novel exercise technology are still scarce to allow for definitive conclusions [28], there is accumulating evidence that WB-EMS may be a viable modality to improve body composition and muscle strength in healthy trained and previously sedentary cohorts [27,29]. Moreover, pioneering studies from our research center and other groups suggest that WB-EMS may also provide beneficial effects in clinical populations, including, for example, improved body composition and physical function in cancer patients [58,59,60] or enhanced oxygen uptake and cardiac function in patients with chronic heart failure [61,62]. Recently, we have demonstrated for the first time that two weekly 20 min WB-EMS sessions during 12 weeks of CR can improve cardiometabolic risk profile in obese women diagnosed with MetS [34].

In the context of this previous research, several findings of the present study were unexpected. First, in contrast to our pilot study [34] and the only other study to date that has examined the effects of WB-EMS during CR [30], our present results do not confirm that WB-EMS effectively maintains SMM over a several-week period of intentional weight loss. Given that previous studies used the same (standard) WB-EMS application (6 s impulse phase, 4 s rest; 85 Hz, 350 μs) [27], these opposite results could be due to differences in participants’ protein intake between the studies. More specifically, the overweight premenopausal women in the study of Willert et al. [30] were supplemented with a multi-component protein powder to achieve a protein intake of 1.2 g/kg/day, and participants in our previous pilot study managed to consume ~1.0 g/kg/day of protein in their calorie-reduced daily diet [34]. In contrast, patients in the present study did not receive any specific protein supplementation in conjunction with WB-EMS exercise, and the majority of participants failed to meet the recommended minimum protein intake during weight loss (≥1.0 g/kg/day) [51], despite detailed advice from a dietitian at study entry and the provision of supporting diet plans. Thus, although still within the recommended dietary allowance for the general population [63], it is conceivable that the amount of ingested protein (~0.8 g/kg/day) was too low to prevent a significant decrease in SMM during CR—despite concomitant WB-EMS exercise. Suboptimal compliance with dietary recommendations is a well-known phenomenon, particularly in obese individuals [64], and recently, the importance of research efforts to develop innovative strategies for achieving higher levels of adherence to a diet has been highlighted [65]. Interestingly, however, SMM was preserved with 1-RT and 3-RT, although protein intakes were also below recommended levels in these both groups, ranging from 0.7 to 0.9 g/kg/day. This observation could indicate that conventional RT elicits a greater anabolic effect on muscle tissue than WB-EMS (at least in obese individuals undergoing CR). However, this assumption remains speculative and requires further investigation.

Second, in contrast to both traditional RT programs, WB-EMS had no significant impact on the cardiometabolic risk profile in our participants, which is also opposite to our pilot study [34]. Previous research on the effects of WB-EMS on cardiometabolic risk outcomes in overweight/obese individuals is still rare, with inconclusive findings. While we [34] and Kemmler et al. [35] found significant improvements in the MetS z-score following several weeks of WB-EMS by an average of 1.2 units, another study reported that WB-EMS did not have positive impact on the overall cardiometabolic risk status [33]. Interestingly, though, the latter study also indicated that, when combined with a specific protein supplement, WB-EMS appeared to induce more favorable changes in the MetS z-score [33]. Given the significant relationship between muscle tissue and cardiometabolic health [13,14,53], this observation supports our assumption that WB-EMS without proper protein intake (either via a protein-rich diet or additional supplements) may possibly not provide a sufficient anabolic stimulus to elicit beneficial effects on SMM and associated cardiometabolic risk factors during CR. Previous studies that have compared the effects of various RT programs with or without additional protein supplementation, however, did not reveal significant differences in terms of changes in cardiometabolic risk outcomes [66,67,68]. Thus, it might be speculated that the efficiency of conventional RT may be less critically affected by suboptimal protein intake compared to WB-EMS, possibly due to its greater anabolic potential.

In contrast to other reports from the literature [8,55], we found only little evidence for treatment effects on blood markers of glucose and lipid metabolism, with the exception of improved cholesterol levels in the 1-RT group and CON group. Since the majority of subjects included in most previous studies were rather overweight to moderately obese subjects [55], this finding could be due to less favorable pre-existing cardiometabolic conditions in our severely obese MetS patients, including altered substrate utilization during exercise [69], potentially associated with worse treatment responses. From a practical point of view, we suggest, therefore, that multimodal exercise programs with additional aerobic training [70] and more targeted nutritional interventions are required for such populations to yield optimal health benefits.

In accordance with previous research investigating the effects of exercise interventions on cardiometabolic health status in obese MetS patients [25,26], the observed MetS z-score improvement following RT was in large part due to a significant reduction in blood pressure. Given that a systolic blood pressure reduction of 10 mmHg has been associated with a lowered risk of CVD and mortality by 20% and 13%, respectively [71], the blood pressure changes in both RT groups are very likely to provide clinically meaningful benefits, comparable to effects achieved with pharmacological antihypertensive treatments [72]. In this context, it was a remarkable finding that, in contrast to traditional RT, blood pressure was not reduced after WB-EMS, but rather tended toward an increase. To date, longer term blood pressure responses to WB-EMS exercise have only been assessed in a few studies with conflicting results. In total, three studies reported significant reductions [33,35,61], one study reported no significant effect [34], and one study reported an increase in blood pressure [62] following WB-EMS. The underlying physiological mechanism responsible for the different effects of WB-EMS and RT on blood pressure and, consequently, on MetS severity in the present study cannot be completely clarified at present, but there could be a possible link to specific adaptations that occur in muscle fibers in response to each type of training.

The human skeletal muscle consists mainly of two muscle fiber types (with several subtypes), which can be distinguished by their contractile properties and metabolic pathways: slow-twitch type 1 fibers and fast-twitch type 2 fibers. Type 1 fibers are characterized by larger amounts of mitochondria and myoglobin, their high oxidative capacity and low fatigability. Type 2 fibers display a smaller mitochondrial content, less myoglobin and lower oxidative capacity but a greater potential for exercise-induced hypertrophy [13,73]. Although fiber type distribution is largely determined by genetic factors [74], the skeletal muscle displays a great plasticity in response to various disease states [13,75] and external stimuli, like exercise [73]. It is well recognized that ageing sarcopenia or cancer cachexia [75], for example, cause a preferential loss of type 2 fibers, while obesity and MetS appear to be associated with type 1 fiber atrophy and a slow-to-fast fiber type shift [76,77]. Additionally, it has been shown that a lower proportion in type 1 fibers is inversely correlated with cardiometabolic indices including blood pressure and arterial elasticity [76,78,79]. Concerning RT, research indicates that when exercise is carried out until failure (as the case in the present study), hypertrophy occurs in both muscle fiber types, independent of load [80]. Scientific consensus on muscle fiber adaptations following WB-EMS, however, is still inconclusive. It has been suggested that muscle fiber recruitment during local electrical stimulation occurs in a non-selective, synchronous way [81], while others found that muscle fiber activation seems to be dependent on the applied stimulation frequency, with frequencies of ≤50 Hz activating mainly type 1 fibers and frequencies of >50 Hz activating more type 2 fibers [82,83]. In line with the latter finding, several authors reported that the standard WB-EMS protocol with 85 Hz (as also applied in present study) appears to be associated with a preferential activation of type 2 fibers [36,83,84,85].

Taken together, it is conceivable that WB-EMS resulted in a more pronounced slow-to-fast fiber type shift compared to RT, which may not be advantageous for improving overall cardiometabolic health in obese MetS patients. However, since no muscle biopsies were taken from the participants during the study, this assumption is hypothetical, suggesting that further studies are required to elucidate the impact of WB-EMS (applied in different frequencies) on muscle fiber adaptations and their relationship to changes in cardiometabolic risk outcomes. Future studies may also wish to investigate more comprehensively the influence of WB-EMS on parameters involved in blood pressure regulation in order to clarify potential adverse effects of electrical stimulation on vascular function.

Besides physical benefits, exercise has also been shown to improve parameters of mental health and well-being [86]. In line with the literature, we observed significant improvements in QoL, as assessed by the EQ-VAS, in both RT groups. Given the reported association between changes in muscular fitness and perceived QoL [87], it is not surprising that such improvements were not evident in the CON group. Moreover, a large multicenter study has documented that positive changes in cardiometabolic health status (in particular blood pressure) are correlated with increases in the EQ-VAS score, which may be attributed to reduced sensation of symptoms and/or awareness of disease improvement [88]. Thus, the lack of improvement in cardiometabolic health status (in particular blood pressure) may explain why QoL remained (consciously or unconsciously) unchanged in the WB-EMS group compared to the RT groups, although we did not find a statistically significant correlation that would confirm this assumption.

Despite the apparent superiority of traditional RT over WB-EMS in improving overall cardiometabolic health status and QoL in obese MetS patients, it should be noted, however, that WB-EMS exercise induced positive effects on body fat percentage and waist circumference, which is (as surrogate measure of abdominal visceral adipose tissue) a central component of MetS [20,37]. More specifically, the average ~3 cm reduction in waist circumference observed in the WB-EMS group represents a change that has been considered clinically meaningful [89]. Moreover, given that muscular strength has been shown to be independently associated morbidity and mortality [90,91], the strength improvements in all major muscle groups following WB-EMS, ranging from 20% to 56%, are very likely to provide distinctive health benefits, including enhanced physical function and orthopedic issues. In line with previous studies in trained and previously untrained populations [27,29], these findings imply that WB-EMS can be considered a solid exercise option to improve body composition (in terms of greater fat loss compared to no exercise) and muscle strength during CR—particularly for individuals with joint problems or other conditions that might limit or prevent participation in conventional RT.

Finally, it needs to be emphasized that no adverse events occurred during the present study, suggesting that both RT and WB-EMS can be safely administered in obese MetS patients, provided that medical clarification is carried out beforehand and that exercise is carried out under close supervision. Particularly in light of the expressed safety concerns regarding the potential risks of WB-EMS-induced muscle damage [92], it is also important to highlight that none of our participants displayed clinically relevant disturbances in CK and creatinine levels after 6 and 12 weeks of exercise, respectively. Furthermore, the high adherence rates (mean value: ~94%) and high ratings of exercise enjoyment (mean value: ~6.1 on a 7-point rating scale) in all three exercise groups indicate that RT and WB-EMS appear to be well tolerated and accepted by obese patients with cardiometabolic disorders.

There are some limitations in our study that should be considered. First, several of our interpretations are based on results obtained from self-reported food records. In this context, it is to note that in general, self-reported outcomes may be associated with potential sources of error, including dishonesty, conscientious responses, or social desirability. During food recording, specifically, it has been reported that individuals rather tend to underestimate their dietary intakes and that the recording per se may (unconsciously) influence the eating behavior [93]. However, it must be pointed out that our participants were highly motivated to change their diet to lose weight and to improve their health condition, and we assume that the careful pre-briefing on how to record food intake should have minimized the extent of potential errors. Furthermore, the reported changes in energy intake agree quite well with the objectively measured participants’ weight loss. Second, we used BIA to determine body composition including SMM, and we are aware that there still exist some concerns regarding this method [94]. However, we note that improvements in technology and the development of segmental multifrequency BIA, as applied in the present study, have substantially enhanced the precision of this method [95]. In particular, the BIA device (seca mBCA 515) used in this study has shown high accuracy for the determination of SMM and body composition compared to the gold standard methods MRI [42] and 4-compartment model [43]. Third, given that the intervention period only lasted 12 weeks, the longer-term effects of the applied exercise protocols remain to be examined in obese individuals at increased cardiometabolic risk. Although our data provide evidence that even small amounts of muscle strengthening exercise in the form of 1-RT can have beneficial health effects on MetS severity, it can be expected that progressively greater training loads will be needed over the longer term in order to continue improving cardiometabolic risk outcomes. Beyond that, it must be taken into account that this study was conducted in a well-controlled setting with careful one-to-one supervision of all exercise sessions. Further research will be required to elucidate whether obese individuals at increased cardiometabolic risk would be able and/or willing to conform with the applied exercise protocols in real-world settings. Such efforts will be important in ensuring the sustainability and public health impact of the exercise protocols used in this study.

## 5. Conclusions

To the best of our knowledge, this was the first study to compare the effects of the increasingly popular exercise method WB-EMS versus traditional RT in a clinical setting. We acknowledge that WB-EMS, due to its time-efficiency and, potentially, the novelty factor per se, may be an attractive option for individuals seeking alternative exercise regimes, which could be helpful to promote physical activity in the population. Beyond that, there is evidence that WB-EMS may also be a promising method to maintain/improve body composition and physical function in patients with accelerated muscle wasting, particularly for those who are not able to participate in conventional RT due to specific contraindications. However, in light of the increasing focus on personalized sports medicine approaches, it is important to prescribe more disease-specific exercise protocols. Thus, the take-home message from this study is that traditional RT (either performed as standard 3-set or single-set training) seems to be more effective on maintaining SMM and improving cardiometabolic health in obese individuals with MetS undergoing CR compared to WB-EMS.

## Figures and Tables

**Figure 1 nutrients-13-01640-f001:**
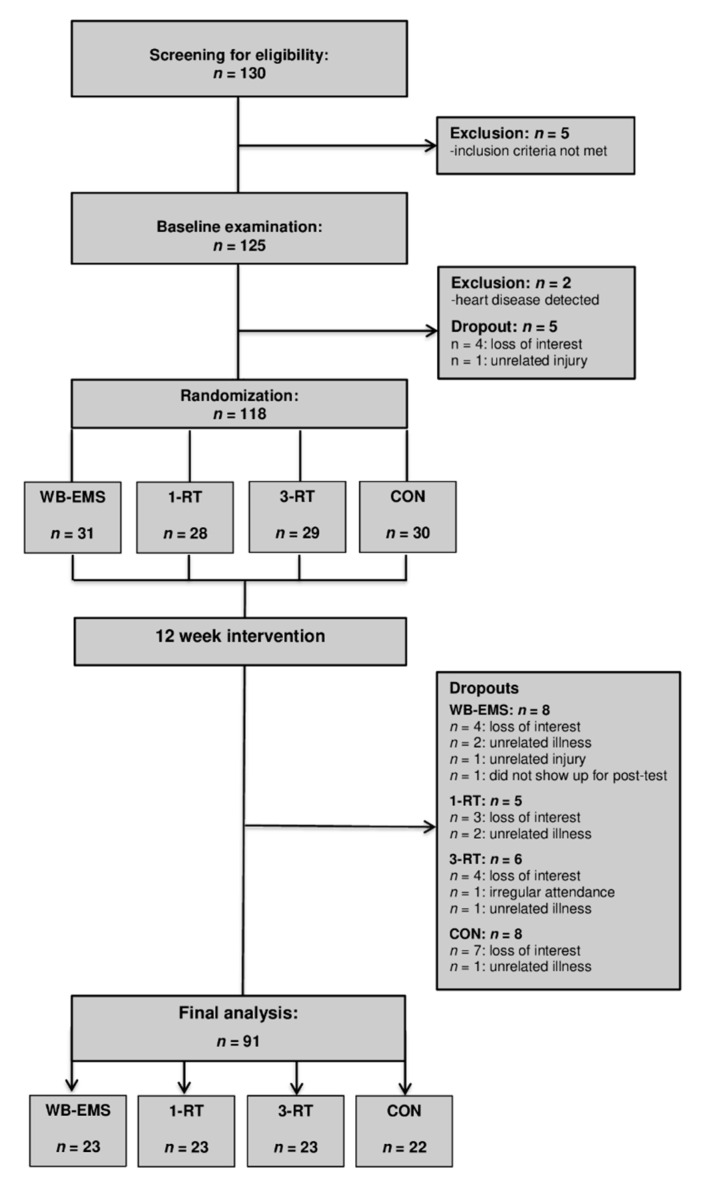
Study flow chart. WB-EMS = whole-body electromyostimulation group, 1-RT = 1-set resistance training group, 3-RT = 3-set resistance training group, CON = control group.

**Figure 2 nutrients-13-01640-f002:**
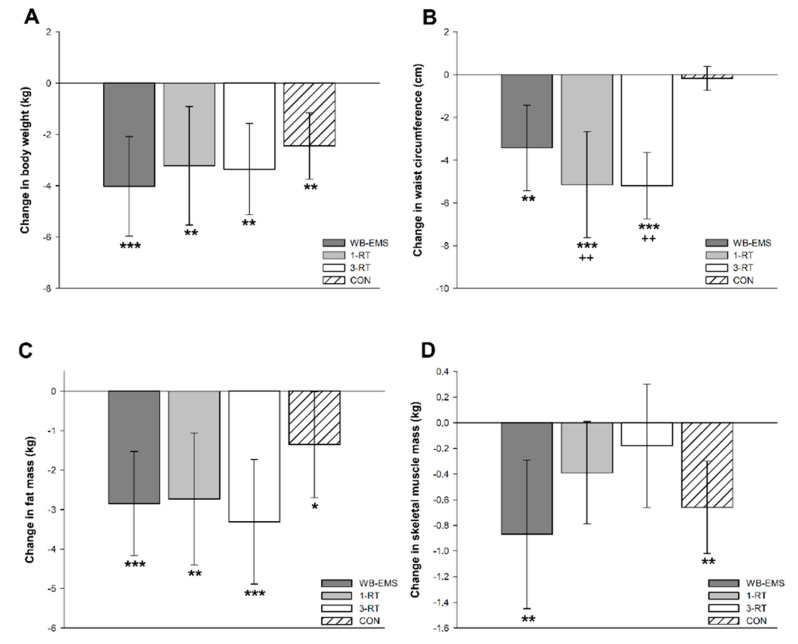
Changes in body weight (**A**), waist circumference (**B**), fat mass (**C**), and skeletal muscle mass (**D**). * (*p* < 0.05), ** (*p* < 0.01), *** (*p* < 0.001): significantly different from pre-intervention.; ++ (*p* < 0.01): significant difference vs. CON.

**Figure 3 nutrients-13-01640-f003:**
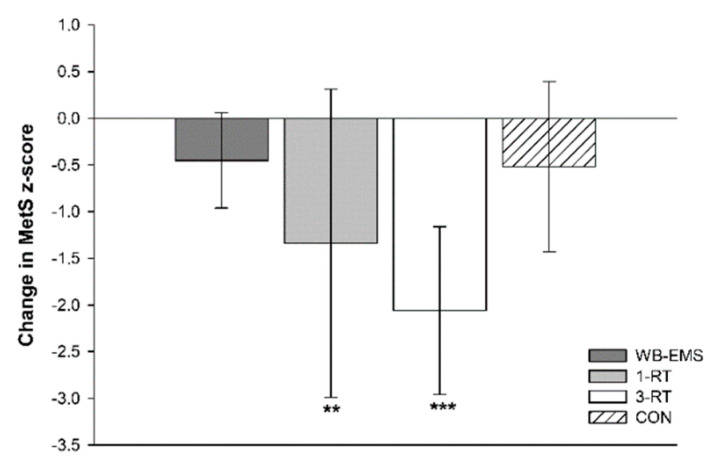
Changes in the metabolic syndrome z-score. ** (*p* < 0.01), *** (*p* < 0.001): significantly different from pre-intervention.

**Figure 4 nutrients-13-01640-f004:**
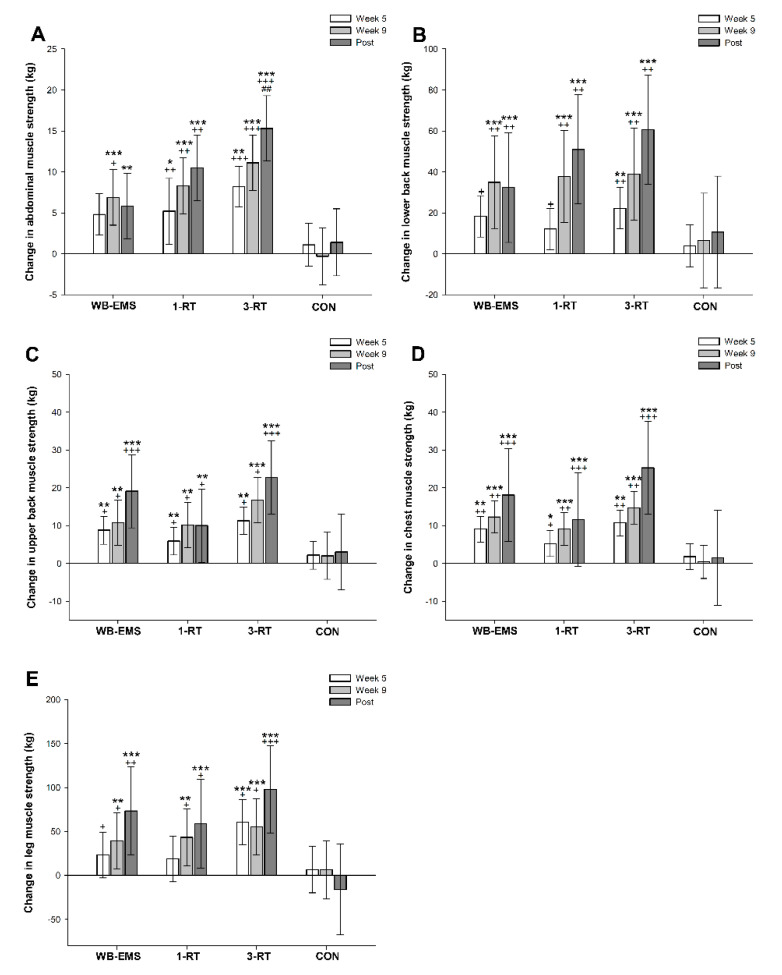
Changes in abdominal (**A**), lower back (**B**), upper back (**C**), chest (**D**), and leg muscle strength (**E**). * (*p* < 0.05), ** (*p* < 0.01), *** (*p* < 0.001): significantly different from pre-intervention.; ^+^ (*p* < 0.05), ^++^ (*p* < 0.01), ^+++^ (*p* < 0.001): significant difference vs. CON. ^##^ (*p* < 0.01) significant difference vs. WB-EMS.

**Table 1 nutrients-13-01640-t001:** Baseline characteristics of all study participants.

Variable	WB-EMS (*n* = 31)	1-RT (*n* = 28)	3-RT (*n* = 29)	CON (*n* = 30)
Gender (f/m)	22/9	20/8	20/8	23/7
Age (years)	51.5 ± 12.1	53.8 ± 12.4	53.9 ± 11.3	51.7 ± 11.7
BMI (kg/m^2^)	37.1 ± 4.3	37.3 ± 7.6	39.8 ± 8.9	38.0 ± 6.3
MetS z-score	2.0 ± 2.6	2.0 ± 4.0	2.8 ± 4.4	2.2 ± 2.6

Values are given as mean ± SD. f = females, m = males, BMI = body mass index, MetS z-score = metabolic syndrome z-score.

**Table 2 nutrients-13-01640-t002:** Daily nutritional intakes pre-intervention and during the last week of intervention.

Variable	WB-EMS (*n* = 20)		1-RT (*n* = 21)		3-RT (*n* = 21)		CON (*n* = 20)	
	Pre	Post	Pre	Post	Pre	Post	Pre	Post
Energy (kcal/d)	2419 ± 663	1922 ± 393 **	2294 ± 792	1789 ± 596 *	2391 ± 714	1876 ± 648 ***	2365 ± 884	1785 ± 811 *
Protein (g/d)	99 ± 33	91 ± 31	100 ± 41	79 ± 26 *	110 ± 41	90 ± 41 *	97 ± 29	71 ± 26 **
Protein (g/kg/d)	1.0 ± 0.3	0.8 ± 0.3	1.0 ± 0.3	0.8 ± 0.2 *	0.9 ± 0.3	0.7 ± 0.3 *	0.9 ± 0.4	0.9 ± 0.4
Fat (g/d)	99 ± 38	79 ± 19 *	96 ± 34	77 ± 32	97 ± 32	76 ± 31 **	103 ± 54	71 ± 44
Fat (g/kg/d)	0.9 ± 0.2	0.8 ± 0.3	0.8 ± 0.3	0.7 ± 0.2 *	1.0 ± 0.5	0.7 ± 0.4 *	0.9 ± 0.4	0.8 ± 0.2 *
CHO (g/d)	250 ± 81	185 ± 58 **	228 ± 81	172 ± 75 **	227 ± 81	179 ± 61 **	237 ± 90	191 ± 81
CHO (g/kg/d)	2.2 ± 0.6	1.7 ± 0.7 **	2.0 ± 0.6	1.6 ± 0.5 **	2.3 ± 1.1	1.9 ± 0.9	2.4 ± 0.8	1.8 ± 0.6 *
Fibres (g/d)	21 ± 10	20 ± 9	23 ± 11	19 ± 8	22 ± 8	20 ± 7	28 ± 15	22 ± 13

Values are given as mean ± SD. CHO = carbohydrates. * (*p* < 0.05), ** (*p* < 0.01), *** (*p* < 0.001): significantly different from pre-intervention.

**Table 3 nutrients-13-01640-t003:** Anthropometric and body composition data pre and post-intervention.

Variable	WB-EMS (*n* = 23)		1-RT (*n* = 23)		3-RT (*n* = 23)		CON (*n* = 22)	
	Pre	Post	Pre	Post	Pre	Post	Pre	Post
Weight (kg)	108.4 ± 19.6	104.4 ± 17.4 ***	104.9 ± 26.1	101.7 ± 25.7 **	117.7 ± 32.3	114.0 ± 32.8 **	104.7 ± 20.2	102.3 ± 20.8 **
BMI (kg/m²)	37.6 ± 4.4	36.3 ± 3.9 ***	37.2 ± 7.6	36.1 ± 7.7 **	40.2 ± 9.1	39.0 ± 9.1 **	37.6 ± 5.8	36.7 ± 5.9 **
FM (kg)	47.6 ± 10.3	44.7 ± 9.2 ***	48.6 ± 16.0	45.8 ± 15.7 **	54.9 ± 19.5	51.6 ± 20.3 ***	47.9 ± 11.6	46.5 ± 12.7 *
FM (%)	44.0 ± 5.0	42.9 ± 5.1 **	45.8 ± 5.7	44.5 ± 6.2 **	46.1 ± 6.9	44.4 ± 7.9 ***	45.9 ± 7.0	45.4 ± 7.7
SMM (kg)	29.7 ± 6.9	28.8 ± 6.3 **	27.2 ± 7.2	26.8 ± 7.3	30.9 ± 9.8	30.7 ± 9.8	27.4 ± 7.6	26.7 ± 7.5 **
TBW (L)	45.4 ± 8.9	44.5 ± 8.1 *	42.3 ± 9.0	41.9 ± 9.2	47.3 ± 12.2	47.2 ± 12.5	42.6 ± 10.0	41.8 ± 9.7 **
Waist (cm)	114.1 ± 10.8	110.8 ± 11.9 ***	111.0 ± 14.3	105.8 ± 12.8 ***	118.3 ± 19.0	113.1 ± 17.7 ***	109.9 ± 11.3	109.8 ± 11.3

Values are given as mean ± SD. BMI = body mass index, FM = fat mass, SMM = skeletal muscle mass, TBW = total body water. * (*p* < 0.05), ** (*p* < 0.01), *** (*p* < 0.001): significantly different from pre-intervention.

**Table 4 nutrients-13-01640-t004:** Cardiometabolic risk variables pre- and post-intervention.

Variable	WB-EMS (*n* = 23)		1-RT (*n* = 23)		3-RT (*n* = 23)		CON (*n* = 22)	
	Pre	Post	Pre	Post	Pre	Post	Pre	Post
MetS z-score	2.33 ± 2.41	1.48 ± 3.10	1.91 ± 3.39	0.57 ± 2.70 **	2.94 ± 4.07	0.88 ± 3.90 ***	2.46 ± 2.92	1.93 ± 2.55
SBP (mmHg)	135 ± 14	137 ± 11	146 ± 18	136 ± 12 *	142 ± 17	132 ± 15 **	134 ± 17	135 ± 13
DBP (mmHg)	86 ± 9	89 ± 10	91 ± 14	85 ± 11	87 ± 9	83 ± 9 **	86 ± 13	85 ± 10
MAB (mmHg)	103 ± 9	105 ± 9	109 ± 13	102 ± 9 **	105 ± 11	99 ± 10 **	102 ± 13	102 ± 10
Glucose (mg/dL)	104 ± 11	103 ± 13	96 ± 15	97 ± 14	102 ± 15	99 ± 14	105 ± 18	101 ± 18
HbA_1c_ (%)	5.7 ± 0.5	5.6 ± 0.5	5.6 ± 0.4	5.5 ± 0.3	5.6 ± 0.3	5.6 ± 0.3	5.7 ± 0.8	5.6 ± 0.6
Triglycerides (mg/dL)	137 ± 67	122 ± 41	124 ± 39	122 ± 36	144 ± 85	126 ± 54	161 ± 72	141 ± 60
Cholesterol (mg/dL)	220 ± 35	213 ± 33	230 ± 29	220 ± 29 *	225 ± 53	217 ± 44	231 ± 45	219 ± 39 *
HDL-C (mg/dL)	54 ± 14	53 ± 13	58 ± 17	57 ± 16	58 ± 17	58 ± 15	53 ± 11	52 ± 12
LDL-C (mg/dL)	145 ± 28	139 ± 25	151 ± 23	142 ± 21 *	143 ± 37	138 ± 33	150 ± 35	146 ± 33
LDL/HDL ratio	2.8 ± 0.9	2.8 ± 0.9	2.8 ± 0.9	2.7 ± 0.8	2.6 ± 0.7	2.5 ± 0.7	2.9 ± 0.8	2.9 ± 0.8

Values are given as mean ± SD. SBP = systolic blood pressure, DBP = diastolic blood pressure, MAB = mean arterial blood pressure, HbA_1c_ = glycosylated hemoglobin A_1c_, HDL-C = high-density lipoprotein cholesterol, LDL-C = low-density lipoprotein cholesterol. * (*p* < 0.05), ** (*p* < 0.01), *** (*p* < 0.001): significantly different from pre-intervention.

**Table 5 nutrients-13-01640-t005:** Muscle strength pre- and post-intervention.

Variable	WB-EMS (*n* = 23)		1-RT (*n* = 23)		3-RT (*n* = 23)		CON (*n* = 22)	
	Pre	Post	Pre	Post	Pre	Post	Pre	Post
Abdominal crunch (kg)	30 ± 15	36 ± 17 **	29 ± 10	39 ± 15 ***^++^	26 ± 12	42 ± 17 ***^+++##^	28 ± 13	30 ± 15
Lower back (kg)	58 ± 32	90 ± 69 ***^++^	64 ± 25	115 ± 89 ***^++^	59 ± 31	120 ± 101 ***^++^	53 ± 25	63 ± 34
Lat pulldown (kg)	43 ± 17	62 ± 47 ***^+++^	39 ± 16	49 ± 20 **^+^	38 ± 17	61 ± 35 ***^+++^	36 ± 16	39 ± 15
Chest press (kg)	32 ± 22	50 ± 54 ***^+++^	30 ± 15	41 ± 19 ***^+++^	28 ± 18	54 ± 51 ***^+++^	28 ± 16	29 ± 18
Leg press (kg)	156 ± 48	229± 30 ***^++^	151 ± 58	209 ± 142 ***^+^	156 ± 95	254 ± 222 ***^+++^	136 ± 50	120 ± 86

Values are given as mean ± SD. ** (*p* < 0.01), *** (*p* < 0.001): significantly different from pre-intervention; ^+^ (*p* < 0.05), ^++^ (*p* < 0.01), ^+++^ (*p* < 0.001): significant difference vs. CON; ^##^ (*p* < 0.01) significant difference vs. WB-EMS.

**Table 6 nutrients-13-01640-t006:** Blood markers of muscle status and renal function at baseline, week 6 and week 12.

Variable	WB-EMS (*n* = 23)	1-RT (*n* = 23)	3-RT (*n* = 23)	CON (*n* = 22)
Creatine kinase (U/L)				
Baseline	181 ± 32	153 ± 33	190 ± 33	166 ± 33
Week 6	210 ± 37	155 ± 37	193 ± 37	148 ± 37
Week 12	215 ± 42	219 ± 43	167 ± 43	131 ± 43
Creatinine (mg/dL)				
Baseline	0.82 ± 0.04	0.79 ± 0.04	0.83 ± 0.04	0.80 ± 0.04
Week 6	0.80 ± 0.04	0.78 ± 0.04	0.83 ± 0.04	0.77 ± 0.04
Week 12	0.80 ± 0.04	0.80 ± 0.04	0.83 ± 0.04	0.78 ± 0.04

Values are given as mean ± SD.

**Table 7 nutrients-13-01640-t007:** Self-reported outcomes pre and post-intervention.

Variable	WB-EMS (*n* = 23)		1-RT (*n* = 23)		3-RT (*n* = 23)		CON (*n* = 22)	
	Pre	Post	Pre	Post	Pre	Post	Pre	Post
EQ5DL5 Index	0.806 ± 0.20	0.836 ± 0.17	0.871 ± 0.12	0.877 ± 0.12	0.820 ± 0.16	0.855 ± 0.12	0.838 ± 0.16	0.730 ± 0.30
EQ VAS (%)	67 ± 20	72 ± 20	67 ± 17	73 ± 16 *	63 ± 18	74 ± 16 ***^+^	57 ± 19	59 ± 21
Exercise enjoyment (0–7)	---	6.2 ± 0.8	---	6.1 ± 0.6	---	5.9 ± 0.8	---	---
Intention to continue with exercise protocol	---	86%	---	95%	---	90%	---	---

Values are given as mean ± SD. * (*p* < 0.05), *** (*p* < 0.001): significantly different from pre-intervention; ^+^ (*p* < 0.05): significant difference vs. CON.

## Data Availability

The datasets generated and analyzed during the current study are not publicly available but are available from the corresponding author on reasonable request.

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
