# Peer review of "Iron Beats Electricity: Resistance Training but Not Whole-Body Electromyostimulation Improves Cardiometabolic Health in Obese Metabolic Syndrome Patients during Caloric Restriction—A Randomized-Controlled Study"

_nutrients, 2021, doi:10.3390/nu13051640_

Round 1

Reviewer 1 Report

I really enjoyed reading this article. It provides a lot of valuable information and highlights the important role of resistance training in the treatment of civilization diseases. Nevertheless, In my opinion, too much attention has been paid to training methods, while very little to nutrition control. I believe that a 3-day control of the dietary records before the experiment and at the end is not enough for people with no experience. Maintaining a deficit for 12 weeks is not easy, especially for people with no experience in dieting or composing meals, etc. I just don’t believe that they were following the diet correctly for 12 weeks, without experience and knowledge about the calorific value of products, they could do it even unintentionally. I do not understand why the authors did not make nutritional counseling for example during the control of the maximal strength test. I think this is a very big limitation.

I have a few more comments:

110: what is the rationale for this hypothesis ”WB-EMS and 1-RT would have similar effects on these outcomes…” 

140-141 did that also apply to supplements? I suspect some participants may have consumed metformin, could this have had an impact on the results?

197: I would suggest not to use the Fmax abbreviation. In experiments of this type, the abbreviation 1RM is commonly used. I understand that this may cause confusion with 1-RT, but in my opinion, Fmax should not be used in this case.

209 did they have any training experience at all? Were the tests for each exercise always performed in the same order?

229: I understand that the participants did not receive specially prepared menus, but they arranged them on their own? Has anyone controlled it? Have they used any app during the experiment to monitor caloric intake? How did the participants control their macronutrients intake? I have great doubts whether the nutrition determined by the participants themselves was correct

2.5.: the training description needs more details. Was the intensity determined relative to the result of the baseline 1RM test? Or each time-point? In addition, how is it possible that the training session time were similar at each intensity as 50% 1RM to muscle fatigue requires about 30 repetitions, while for 80% 1RM it is ~ 8 repetitions. The 1RM test was also performed at such tempo of movement?

402: Table 5?

467-468 - I would consider changing that sentence. The maximum strength only decreased in the case of leg press, however, in the rest of the exercises, insignificantly but improved.

574 - 605 is this paragraph really necessary? Is it consistent with the purpose of the study?

Author Response

Comments and Suggestions for Authors

I really enjoyed reading this article. It provides a lot of valuable information and highlights the important role of resistance training in the treatment of civilization diseases. Nevertheless, In my opinion, too much attention has been paid to training methods, while very little to nutrition control. I believe that a 3-day control of the dietary records before the experiment and at the end is not enough for people with no experience. Maintaining a deficit for 12 weeks is not easy, especially for people with no experience in dieting or composing meals, etc. I just don’t believe that they were following the diet correctly for 12 weeks, without experience and knowledge about the calorific value of products, they could do it even unintentionally. I do not understand why the authors did not make nutritional counseling for example during the control of the maximal strength test. I think this is a very big limitation.

Response:

Dear reviewer,

First of all, we would like to thank you for your time and effort taken to review our manuscript.

We are very pleased that you have enjoyed reading our paper and we appreciate your valuable comments and suggestions on how to improve the document. Please find our point-by-point responses to your comments below. We have made revisions according to your comments and suggestions, wherever possible. All changes in the manuscript are highlighted in yellow.

Regarding your first (major) comment:

Indeed, the major focus of our study was to compare the effects of the different resistance training modalities on cardiometabolic health and several other outcomes, while the nutritional intervention (caloric restriction) was applied as a “standard care treatment” according to international obesity guidelines.

With respect to your concerns, we would like to note the following points:

First, all participants (100%) of the present study had reported that they have already tried to reduce their body weight in the past years with various diets. They also reported that they had several times discontinued attempts to reduce their weight in frustration. In sum, due to (unsuccessful) participation in numerous previous weight loss programs, all participants had already extensive experience in dieting and a surprisingly profound knowledge about composing meals and food recording. In contrast, the vast majority of participants had no experience with exercise, in particular resistance training.

Accordingly, we have added this information to the Results Section, please see: p. 7, lines 300-301.

“All participants had already experienced multiple weight loss attempts in the past.”

Second, the baseline 3-day food recording was extensively evaluated by the dietitian and discussed together with the participants to ensure a proper documentation of caloric and macronutrient intakes. Subsequently, after individual calculation of the daily calorie and macronutrient needs, participants received detailed food lists with corresponding calorie amounts and receipts.

Third, it also important to highlight that apart from the nutritional counseling conducted at study entry and the detailed written information handed out to all participants (as already stated in the previous version of the manuscript; revised version, please see: p. 5, lines 235-236), our nutrition team (consisting of 3 dietitians, 3 nutrition nurses and 1 nutrition scientist) was available throughout the whole study period to answer any upcoming nutrition questions. However, as described above, given that all participants had already years of experience with diets, this additional offer for consultation was only very scarcely used.

We have also added this information to the text. Please see: p. 5, lines 235-238.

“[…] and food list with corresponding calorie amounts were provided to support participants on how to implement the dietary recommendations at home. Additional consultation was offered by our nutrition team throughout the whole study period if questions arose regarding the diet.”

Regardless of these points, we are of course well aware that self-reported outcomes (including food recording) are associated with several potential sources of error, which we have already discussed in the Limitation Section of the manuscript. Revised version, please see: Discussion, p. 16, lines 647-656.

We hope that these explanations were helpful to clarify your concerns and will drive home this critical point.

I have a few more comments:

Comment 2:

110: what is the rationale for this hypothesis ”WB-EMS and 1-RT would have similar effects on these outcomes…”

Response:

The rationale for this hypothesis was mainly based on previous research by Kemmler et al. (2016, references 35,36), which has been the only study to date to directly compare WB-EMS vs. conventional weight-machine-based RT. Similar to our study (with small differences), RT was also applied as a single-set-to-failure protocol after an initial preparation period in this investigation.

As already stated in the previous version of our manuscript (revised version, please see p. 3, lines 99-102), the authors of this study found that both exercise modalities, which they classified “comparably time-efficient”, induced similar improvements in body composition, strength and cardiometabolic risk outcomes in healthy untrained men.

According to your concern, we have added one more piece of information to our hypothesis to make this point more salient. Please see: p. 3, lines 113-114.

“[…] WB-EMS and 1-RT would have similar effects on these outcomes, as observed in previous research in healthy untrained men [35,36], […]”

Comment 3:

140-141 did that also apply to supplements? I suspect some participants may have consumed metformin, could this have had an impact on the results?

Response:

You raise an important point. Only a negligible number of participants (n=3) reported that they had used trace element and vitamin supplements (n=1: magnesium; n=1: multivitamins, n=1: Vit D). However, these supplements were already taken for more than 6 months prior to study enrollment and the amount was not changed during the study.

We have added this information. Please see p. 3, line 144.

“Participants agreed not to change their medications/supplements […]”

Moreover, it is important to note: None of the participants reported to use specific supplements that have a significant impact on muscle growth, such as protein, BCAA’s, creatine or similar supplements.

Metformin was used by n=8 participants (including 4 dropouts). As with supplements, the dosage was stable for more than 6 months prior to study enrollment and no changes were done during the study.

Comment 4:

197: I would suggest not to use the Fmax abbreviation. In experiments of this type, the abbreviation 1RM is commonly used. I understand that this may cause confusion with 1-RT, but in my opinion, Fmax should not be used in this case.

Response:

Thank you for this suggestion. Accordingly, we have changed the terminology into “1RM” throughout the whole manuscript.

Comment 5:

209 did they have any training experience at all? Were the tests for each exercise always performed in the same order?

Response:

Thank you for drawing our attention to this.

Apart from a very small number (n= 3) of male participants, who had performed resistance training for a short period more than at least 3 years ago, previous exercise experiences of participants were very rare and limited to some types of aerobic exercise. All participants reported a sedentary lifestyle including no specific exercise for at least 1 year prior to the study. We agree that the wording may be misleading and thus, we have changed the sentence accordingly. Please see: p. 5, lines 210-211.

“[…] based on the therapist’s experience and participants’ feedback on the chosen weight load.”

Regarding your second comment:

All tests were performed in a standardized (same) order.

We have added this information. Please see: p. 5, line 207.

“[…] participants performed 1RM tests on the following five devices in a standardized order.”

Comment 6:

229: I understand that the participants did not receive specially prepared menus, but they arranged them on their own? Has anyone controlled it? Have they used any app during the experiment to monitor caloric intake? How did the participants control their macronutrients intake? I have great doubts whether the nutrition determined by the participants themselves was correct

Response:

Thanks again, for your suggestion. Yes, participants arranged their menus on their own, which represents a real-world scenario.

We did not use any App to monitor caloric and macronutrient intake. We agree that Apps may be helpful in some specific cases to monitor caloric intake. However, in our study it was difficult to implement an App due to several reasons, as for example: the participants used many different types of cell phones; different affinities of participants to the use of electronic devices; compliance with data protection when using commercial apps.

Furthermore, please see our previous responses to your comment #1.

Comment 7:

2.5.: the training description needs more details. Was the intensity determined relative to the result of the baseline 1RM test? Or each time-point? In addition, how is it possible that the training session time were similar at each intensity as 50% 1RM to muscle fatigue requires about 30 repetitions, while for 80% 1RM it is ~ 8 repetitions. The 1RM test was also performed at such tempo of movement?

Response:

Thank you for your suggestions.

First, intensity was initially determined relative to the result of the baseline 1RM test and then (as previously stated in the first version of the manuscript; revised version please see: p. 6, line 269-270) adjusted according to follow-up 1RM tests (every 4 weeks) to account for training progression. As stated in the text, all sets were performed until failure.

Second, you are absolutely right. Thank you for this important suggestion. Depending on the intensity and the corresponding higher/lower number of repetitions, total session time decreased during the study (shorter sessions times at higher intensities). We have revised this section in the text. Please see: p. 6, lines 276-278.

“As the number of repetitions decreased with increasing intensity, the mean session time decreased during the training period from ~20 min/session to ~11 min/session in the 1-RT group and from ~60 min/session to ~38 min/session in the 3-RT group, respectively.”

Third, yes, exactly. For standardization purposes, the 1RM test was performed with the similar tempo of movement.

Comment 8:

402: Table 5?

Response:

Thank you for drawing our attention to this. We apologize for the typo, which we have corrected.

Please see: p. 10, line 407.

“Pre- and post-intervention values are shown in Table 5 […]”

Comment 9:

467-468 - I would consider changing that sentence. The maximum strength only decreased in the case of leg press, however, in the rest of the exercises, insignificantly but improved.

Response:

According to your suggestion, we have revised the sentence. Thank you. Please see p. 13, lines 473-474.

“Accordingly, the CON group did not experience significant improvements in overall cardiometabolic health status and muscle strength the in post-intervention examinations.”

Comment 10:

574 - 605 is this paragraph really necessary? Is it consistent with the purpose of the study?

Response:

We understand your concern and agree that the paragraph might go into great detail on this specific subject. However, in our opinion, the unexpected difference between WB-EMS and traditional RT regarding their effects on blood pressure changes is a crucial finding of our study and we believe that the information we provide in this section will be helpful for the reader and stimulate a lively scientific discussion. We very much hope for your appreciation that we would like to keep this section in the manuscript. Thank you very much.

Reviewer 2 Report

The present study sought to determine the impact of whole-body EMS on cardiometabolic health, specifically MetS, as well as whole-body EMS and various resistance training protocols during caloric restriction on MetS, body composition, and other quality of life outcomes. The main findings of the study are that whole-body EMS did not have the predicted effects on MetS z-score nor did it attenuate SMM loss or lead to improvements in quality of life scores during caloric restriction, whereas the resistance training groups did see a sparing of SMM loss and did improve MetS z-score and quality of life scores.  

I see very little to edit here. The study was conducted soundly, the results presented clearly, and the findings were not overstated. The limitations of the study were fully illuminated in the discussion by the authors. 

Author Response

Dear reviewer,

Thank you very much for your kind comments and for recommending our manuscrit for publication. We appreciate it a lot.

Best wishes!

Round 2

Reviewer 1 Report

In my opinion, the authors have well addressed all my concerns and suggestions. I believe that the article deserves to be published in Nutrients.